# Spatiotemporal Changes in the Watershed Ecosystem Services Supply and Demand Relationships in the Eastern Margin of the Qinghai-Tibetan Plateau

Yuehua Zhu, Yixu Wang, Zongdong Hou, Jing Shi and Jie Gong *

Key Laboratory of Western China's Environmental Systems (Ministry of Education),
College of Earth and Environmental Sciences, Lanzhou University, Lanzhou 730000, China
* Correspondence: jgong@lzu.edu.cn; Tel.: +86-138-932-56119

**Abstract:** Clarifying the spatiotemporal changes in the supply and demand relationship of ecosystem services (ESs) is essential for optimizing ESs management. However, several studies have reported the ESs supply and demand risk in complex mountainous areas. In this study, we quantitatively analyzed the spatiotemporal variation in ESs supply, demand, and their trade-off and synergy, including water yield, soil conservation, and food provision in the Bailongjiang watershed (BLJW) in western China. The results showed that the total supply and demand of water-yield and soil-conservation services rose with a surplus from 2002 to 2018, except for food provision. A high value characterizes the water-yield and soil-conservation supply in the south, but there are low values in the east BLJW. The spatial distribution of water and food supply–demand featured a high demand in the subareas with population aggregation. Soil-conservation demand is high in the northwest and south of Wudu. The dominant spatial matching type of supply and demand in water yield was a high supply with a low demand. Soil conservation was associated with a low supply and low demand, and food provision with a high supply and increased demand. A synergy existed between water yield and soil conservation. Trade-offs existed between water yield, food provision, and soil conservation. The spatial distribution of trade-off intensity showed distinctive patterns. The supply–demand ratio of WY and SC decreased with the increasing trade-off intensity. This study comprehensively considers ES and supply–demand conflicts, thus providing a new perspective and approach for enhancing ecosystem services and high-quality regional development.

**Keywords:** ecosystem services supply and demand; match and mismatch; spatiotemporal changes; trade-offs and synergies; the Qinghai-Tibetan Plateau



## 1. Introduction

Ecosystem services (ESs, hereafter) refer to the benefits that humans obtain directly or indirectly from the ecosystem to satisfy and maintain all the necessities of survival and development [1,2]. As a link between the natural and social–economic systems, the changes in ESs are closely related to human well-being. The linkage and interaction are mainly reflected by the relationship between ESs supply and demand [3]. ESs supply refers to the ability the ecosystem produces for humans [4,5], and ESs demand refers to the sum of ESs products humans consume and use [6]. The natural ecological and social systems are always mixed as a coupled system in which ecosystem structures and processes cannot form ESs without human beings as beneficiaries [7]. Thus, study on ESs supply or demand will inevitably benefit the effective management of ESs, prevent ESs degradation, maintain sustainable supply, and improve human well-being [8]. Currently, the research content on ESs is bursting, a new comprehensive framework of "ESs supply—ESs flow—ESs demand—human well-being improvement" is being formed and improved gradually [9], and the ESs supply and demand have become one of the new highlights. Earlier research on the ESs supply and demand focused more on the ecological carrying capacity and

monetary evaluation of ESs [10–12], followed by the quantitative human consumption of natural resources, such as the quantification of ESs supply and demand [13,14]. With the continuous enrichment of research content, the study topics have been further expanded to include quantity and space matching [15–17], the balance evaluation of different scales [18], trade-off/synergy relationships [19,20], flow path [21–23], influence mechanisms [24,25], and the ecological management of ESs supply and demand [26,27], etc.

Exploring the quantitative relationship, and spatial match/mismatch between ESs supply and demand can help us to comprehensively and effectively understand the spatial allocation of natural resources [28–30]. Current research methods which are used to quantify the ESs supply and demand mainly include the value assessment, participation, and ecological model methods [31]. The value assessment method primarily uses the ESs value per unit or monetary valuation (e.g., market value and cost avoidance) to estimate the ESs supply and demand value based on the method proposed by Costanza et al. [32], which has been used worldwide at the national and continental levels with an intuitive expression. However, the accuracy is insufficient regarding local differences within the study area [33–35]. The participation method uses the expert evaluation matrix and questionnaire to evaluate ESs supply and demand, which is more suitable for capturing ES demand but is highly subjective and susceptible to the personal factors of experts and respondents, uses high investigation costs, and mainly applies to the small and medium scales [36–39]. Different ESs models have been proposed and used widely based on various ecological theories. Most of the ecological models concerning the ESs are mainly focused on the supply or demand side and lack matching assessment methods/models (e.g., the social values for ecosystem services (SolVES) and the integrated valuation of ecosystem services and trade-offs (InVEST) model) [19,40,41]. Models such as ARIES (artificial intelligence for ES) can estimate the ESs supply and demand, but it is difficult to obtain some local parameters, which impedes the widespread use of the model [42]. Therefore, it is essential to build corresponding models to match the calculation methods and consistency dimensions to further explore the supply and demand relationship. The relationships between ESs refer to the interaction and interrelation between different ESs, including trade-offs, synergies, and no-effects [43,44]. In order to reduce the contradiction between the trade-offs of ESs supply or demand and to balance the relationships between ESs supply and demand, it is vital to understand the characteristics of the ESs relationships at different times and spaces to promote the sustainable management of natural capital and ESs [45–47]. Current methods of ESs supply–demand relationship quantification mainly include spatial mapping to directly map or overlay the ESs supply and demand [48–50], quantitative analysis of the supply–demand relationship based on statistical methods [51–53], and simulation analysis of different scenarios [54,55]. The comprehensive utilization of multiple ESs with complementary advantages will help in better understanding the ESs supply and demand relationship.

The Bailongjiang watershed (BLJW, hereafter) in the eastern margin of the Qinghai-Tibetan Plateau has critical ecological functions of water and soil conservation [56]. Preliminary investigations show that the BLJW urgently needs to improve food production and soil and water protection [57,58]. Thus, the research prioritizes water-yield, soil-conservation, and food-provision services as the crucial ESs in the BLJW. Certain relevant studies have shown that the mismatch between ESs supply and demand is a potential cause of ecosystem degradation [59–61]. A comprehensive analysis of the ESs supply, demand, and their relationships is vital for decision-makers to identify the potential risk areas that need priority attention due to the supply shortage [62–64]. There are many studies on the ESs supply trade-offs or the mismatch between ESs supply and demand. However, the comprehensive studies on both are still mainly focused on the theoretical framework, lacking practical case studies [65,66]. Research shows that there is also an inherent relationship between ESs supply–demand dynamics and trade-offs [67]. For example, the enhancement of an ES supply is at the cost of reducing another ES in the trade-off relationship, which may lead to an increase in the contradiction between the ES supply and demand. Accordingly, the

human need may reduce the supply of other ESs and affect the trade-off relationship. Thus, analyzing the coupling of ESs supply–demand dynamics and trade-off characteristics will intensify the ESs theory and provide solutions to alleviate the conflicts between ESs supply and demand.

The aim of this study is to reveal the spatiotemporal changes of ESs and their supply and demand relationship for countermeasures and suggestions to enhance ESs and ecological management. Specifically, the research aims include: (i) to analyze the spatiotemporal variation in water yield, soil conservation, food provision, and their supply and demand; (ii) to reveal the quantitative and spatial matching relationship between ESs supply and demand, as well as the trade-offs and synergies between ESs; (iii) to explore the internal relationship between the supply and demand relationships of ESs and the ESs trade-off/synergy, and the influencing factors; and (iv) to put forward suggestions on ecosystem management to improve ESs and alleviate the supply and demand conflict of ESs in the BLJW.

## 2. Study Area and Methods

### 2.1. Study Area

The Bailongjiang watershed (BLJW) (32°36′–34°24′ N, 103°00′–106°30′ E), with a total area of approximately 18,400 km², is located in the transitional ecotone from the Qinghai-Tibetan Plateau to the western Qinling Mountains and the Loess Plateau (Figure 1). The elevation in the BLJW is high in the northwest and low in the southeast (568–4860 m). The geomorphic types are mainly plateau mountains, river valleys, and loess hills with complex topographical and geological conditions [68]. The average yearly temperature is 6–15 °C, and the average precipitation is 500–900 mm, both showing a decreasing trend from southeast to northwest [69,70]. Significant spatial differences exist in human activities and economic development in the BLJW. The upstream areas mainly engage in forestry and animal husbandry, while the downstream areas are primarily engaged in agriculture. The most densely settled regions are concentrated in the downstream areas of the Minjiang, the eastern part of the Zhouqu-Wudu section of the Bailongjiang, and the banks of the Baishuijiang. As of 2018, the total population of the whole watershed is about 1.2 million, of which the Wudu account for about 48%. Due to the intensive human activities and complex natural environment, the fragile ecology is one of the high-incidence places experiencing frequent geo-disasters such as debris flow and landslides in China. Therefore, coordinating the supply and demand of ESs is vital for ecological protection and human welfare.

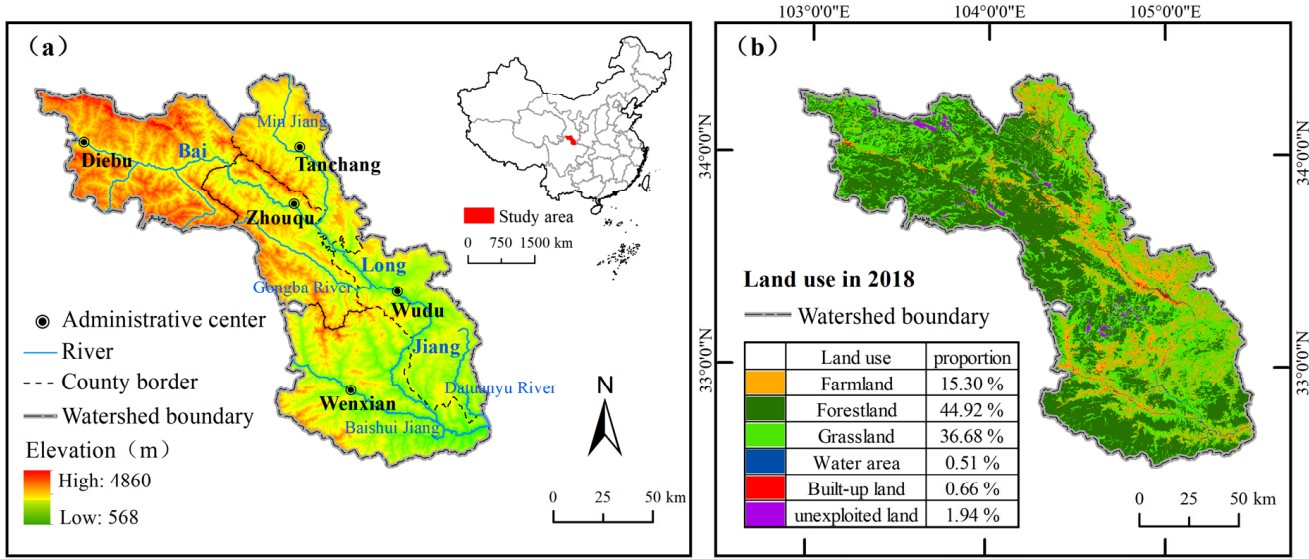

**Figure 1.** The position (**a**) and land use (**b**) of study area.

*2.2. Research Framework*

The primary purpose of this study is to propose an analysis framework for integrating the ESs supply and demand relationships and the ESs trade-offs and synergies, to understand the supply and demand spatial distribution and dynamic relationship of key ESs in the BLJW on the eastern edge of the Qinghai Tibet Plateau, and to propose countermeasures and suggestions to achieve the balanced spatial allocation of ecological and environmental resources (Figure 2).

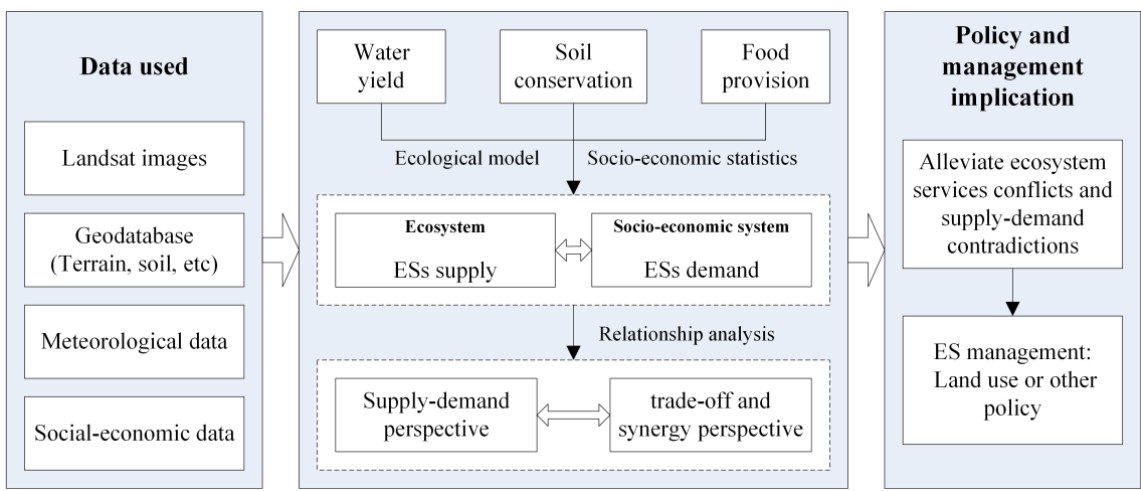

**Figure 2.** The overall research framework of this study.

*2.3. Data Sources and Processing*

The data source of this study refers to the existing research around the BLJW [56,58,68]. The remote-sensing image data were obtained from the United States Geological Survey "https://glovis.usgs.gov/ (accessed on 21 August 2020)" with a resolution of 30 m in August 2002, 2010, and 2018, respectively. The preprocessing method mainly included radiometric correction, geometric correction, band fusion, image mosaic, and cropping. The land-use data were obtained from Landsat TM images via ENVI preprocessing and manual visual interpretation, combined with Google Earth and field validation, with an estimated accuracy of over 86%. According to the LUCC classification standard and the actual situation of the BLJW, the land use in the study area was divided into six categories: farmland, forestland, grassland, water area (lakes, rivers, reservoirs, etc.), built-up land (residential, industrial and mining land, towns, etc.), and unexploited land (bare rock, barren beach, sandy land, mountain snow, etc.). The DEM data with a 30-m resolution were obtained from ASTER GDEM data "http://www.gscloud.cn (accessed on 24 September 2021)". The NDVI dataset was obtained from MODIS13Q1 products "https://ladsweb.modaps.eosdis.nasa.gov/ (accessed on 29 July 2020)" with a spatial resolution of 250 m and a temporal resolution of 16 days. Using the ArcGIS spatial analysis module for annual maximum analysis, splicing, clipping, and resampling, 30-m NDVI data of the BLJW from 2002 to 2018 were obtained. The meteorological data of temperature, precipitation, and solar radiation were derived from the daily data during 2000–2020 of the 17 meteorological stations "http://data.cma.cn (accessed on 9 June 2021)" in and surrounding the BLJW. ANUSPLIN was used for the spatial interpolation of meteorological data, and the spatial resolution was set at 30 m × 30 m. The soil data used in this study were clipped and rasterized from the 1:1 million soil-type data of China "http://www.resdc.cn/ (accessed on 13 March 2021)" and the second soil census data of Gansu Province. The soil data were verified by relevant literature in the China Soil Database "http://vdb3.soil. csdb.cn/ (accessed on 6 August 2021)", field sampling, and experimental analysis. The socio-economic statistics data were obtained from Gansu Province and county statistical

yearbooks, water conservancy statistical yearbooks, and relevant government work reports "http://tjj.gansu.gov.cn/ (accessed on 22 October 2022)".

*2.4. Quantification of ESs Supply and Demand*

Based on the typical natural conditions and socio-economic characteristics of the BLJW, we selected water yield (WY), soil conservation (SC), and food provision (FP) to reveal the spatiotemporal changes of the ESs and their supply and demand relationships.

2.4.1. Water Yield

The simplified water-yield model in InVEST was used to estimate the water supply (*WS*) [71]. The water demand (*WD*) was defined as the summed quantities of domestic water, industrial water, and farmland-irrigation water by the water-quota method [72]. The expressions are as follows:

$$WS_x = \left(1 - \frac{AET_x}{P_x}\right) \times P_x \tag{1}$$

$$WD_x = D_{perwater} \times P_{xpop} \tag{2}$$

where $WS_x$ is the water-yield supply (mm); $P_x$ is the average annual precipitation (mm); $AET_x$ is the actual yearly evapotranspiration (mm); and the detailed model parameters are from previous research results [68,73,74]. $WD_x$ is the water-yield demand (m³); $D_{perwater}$ is the water demand per capita (m³); $P_{xpop}$ is the population density (per km²) [75]; $x$ is a random pixel.

2.4.2. Soil Conservation

The soil-conservation supply and demand (*SS* and *SD*) are estimated based on the revised universal-soil-loss equation by using the sediment delivery ratio module in InVEST [58,71,76]. The expressions are as follows:

$$SS_x = RKLS_x - USLE_x + SEDR_x \tag{3}$$

$$SD_x = USLE_x = R_x \times K_x \times LS_x \times C_x \times P_x \tag{4}$$

where $SS_x$ is the soil-conservation supply; $RKLS_x$ is the potential erosion of soil; $SEDR_x$ is the sediment retention; $USLE_x$ is the actual erosion of soil; $SD_x$ is the soil-conservation demand; $R_x$, $K_x$, $LS_x$, $C_x$, and $P_x$ are precipitation erosivity factor, soil-erodibility factor, slope-length gradient factor, vegetation-coverage factor, and support-practice factor, respectively. The model parameters refer to previous studies [56].

2.4.3. Food Provision

There is a significant linear relationship between farmland NDVI and food production [76]. Therefore, the cultivation-area NDVI is used to calculate the food supply (*FS*). The various kinds of food are converted into energy values to avoid dimensional differences between different foods, according to the China Food Composition 2004 (Volume 2). The food demand (*FD*) is estimated according to the per capita food demand multiplied by the population [19,77]. The expressions are as follows:

$$FS_x = \frac{NDVI_{x,j}}{NDVI_{sumj}} \times c_{sumj} \tag{5}$$

$$FD = P_{pop} \times c_{avg} \tag{6}$$

where $FS_x$ is the food supply (KJ); $NDVI_{x,j}$ is the NDVI of the corresponding land type *j* on grid *x*; $NDVI_{sumj}$ is the sum of NDVI on land *j*; $c_{sumj}$ is the total output of land type *j* (KJ);

*FD* is the food demand (KJ); $P_{pop}$ is the population density (per km$^2$); $c_{avg}$ is the per capita food demand (KJ).

### 2.5. Relationship between ESs Supply and Demand

This study used the ecological supply–demand ratio (*ESDR*) to depict the quantitative relationship between ESs supply and demand [78,79]. A positive value of *ESDR* indicates a surplus in ESs, a zero value indicates a balance between supply and demand, and a negative value indicates a deficit in ESs. The bivariate Moran's I indicator was used to depict the spatial-matching relationship of ESs supply and demand [80]. Based on the Z-test ($p < 0.05$), the local indicators of spatial association (*LISA*) map was used to reflect five types of spatial-clustering relationships: not significant, high–high (high supply–high demand), low–low (low supply–low demand), high–low (high supply–low demand), and low–high (low supply–high demand). The expressions are as follows:

$$ESDR = \frac{ESS - ESD}{(ESS_{\max} + ESD_{\max})/2} \tag{7}$$

$$LISA_i = \frac{1}{n} \frac{(x_i - \overline{x})}{\sum_i (x_i - \overline{x})^2} \sum_j w_{ij}(x_i - \overline{x}) \tag{8}$$

where *ESS* and *ESD* are ESs supply and demand, respectively; $ESS_{max}$ and $ESD_{max}$ are the maximum values of supply and demand of the selected service; $x_i$ is the attribute value of the spatial unit $i$; $\overline{x}$ is the mean attribute value; $n$ is the total number of spatial units; $w_{ij}$ is the spatial weight matrix between the spatial units $i$ and $j$.

### 2.6. ESs Supply and Demand Trade-Off and Synergy

This study used correlation analysis to reveal the trade-off and synergistic relationship between the supply and demand sides of ESs. When the correlation value is positive and passes the significance test, it indicates that the ESs pair is a synergetic relationship; when the correlation value is negative and passes the significance test, it suggests that the service pair is a trade-off relationship; if the significance test is not passed, the relationship is not apparent. In order to further validate and quantify the relationships among ESs and spatial visualization, the root mean standard deviation (*RMSD*) [81–83] was applied to reveal the trade-off and synergy degree among the three ESs. The expression is as follows:

$$RMSD = \sqrt{\frac{1}{n-1} \times \sum_{i=1}^{n} \left(ES_{st} - \overline{ES_{st}}\right)^2} \tag{9}$$

where *RMSD* is the trade-off intensity; $ES_{st}$ is the standardized value of the selected ES; $\overline{ES_{st}}$ is the mean value of the n ESs; $n$ indicates the number of ESs. The essence of the *RMSD* is to calculate the distance between the coordinate points of the two ecosystem services after standardization and the 1:1 line.

## 3. Results and Analysis

### 3.1. Spatial Distribution of the ESs Supply and Demand

Except for the food-provision demand, the total ESs supply and demand increased from 2002 to 2018 in the BLJW (Table 1). A distinctive pattern of the spatial distribution of ESs supply and demand was observed (Figure 3). The high-value areas of water-yield supply were mainly in south Wenxian and southeast Wudu (water-yield supply is above 5000 m$^3$/hm$^2$) with a high vegetation coverage and annual precipitation. The median-value areas of the water supply were mainly distributed in the northwest BLJW, with a high altitude, low temperature, and high evaporation, while the low-value areas were distributed in the densely populated area in the north and the adjacent regions of the Bailongjiang, such as the Tanchang-Zhouqu-Wudu section of the BLJW (Figure 3A). Due

to the population concentration and developed industry, the water-yield demand was concentrated in the residential and industrial areas around Wudu and near the river-bank corridors. The low-value water-demand areas were distributed in the forest, mountain, rocky, and unexploited lands with a sparse population in the BLJW (Figure 3D). From 2002 to 2018, the water supply in the BLJW increased from 3304.55 $m^3/hm^2$ to 4419.92 $m^3/hm^2$, and the water demand increased from 117.91 $m^3/hm^2$ to 276.03 $m^3/hm^2$ (Table 1). The water demand in Wudu increased significantly and the total water supply far exceeded the total demand. The soil-conservation supply and demand in the BLJW were generally low. As the region contains complex topographical and geological conditions, the distribution of high-value supply and demand areas of soil conservation was fragmented (Figure 3B,E). The northwest of the BLJW has a high slope and high potential soil erosion. The southeast of the BLJW has a high potential precipitation erosion. Due to the high forest coverage in the northwest and southeast BLJW, the actual soil-conservation supply amount is relatively high. High demand for soil conservation is mainly concentrated in the subareas with reasonable soil-conservation measures. From 2002 to 2018, soil-conservation supply increased from 192.10 $t/hm^2$ to 520.58 $t/hm^2$, and soil-conservation demand increased from 27.12 $t/hm^2$ to 69.63 $t/hm^2$. Despite the substantial quantitative difference between soil-conservation supply and demand amounts, the *ESDR* was merely 0.05, and supply and demand were balanced (Table 1). The high-value areas of food supply are mainly located in the agricultural areas with good land resources in Zhouqu, Wudu, and Wenxian. Most of the BLJW belongs to low-value subareas because of the fragmentation of the mountainous terrain. The spatial distribution of food-provision demand is closely related to the distribution of population density. The population of the BLJW is concentrated in the eastern parts (Figure 3C,F). The food supply nearly doubled from 2002 to 2018, while the demand decreased. According to the food-demand data in the statistical yearbook of each county in the watershed, the demand of residents for high-energy food (wheat, etc.) decreased. In contrast, the demand for low-energy food (vegetables, etc.) increased, and the total food-energy demand decreased by 23.86%. The total food supply was low in 2002 and 2010 and oversupplied in 2018, but the average *ESDR* of the BLJW was less than zero (Table 1).

**Table 1.** ESs supply and demand in the BLJW from 2002 to 2018.

| ESs | ESs Supply | | | ESs Demand | | | *ESDR* | | |
|---|---|---|---|---|---|---|---|---|---|
| | **2002** | **2010** | **2018** | **2002** | **2010** | **2018** | **2002** | **2010** | **2018** |
| Water yield ($\times 10^9\, m^3$) | 6.03 | 6.49 | 8.07 | 0.21 | 0.41 | 0.49 | 0.91 | 0.90 | 1.07 |
| Soil conservation ($\times 10^8$ t) | 4.53 | 7.73 | 9.18 | 0.41 | 0.85 | 1.02 | 0.05 | 0.05 | 0.05 |
| Food provision ($\times 10^6$ MKJ) | 2.43 | 4.10 | 4.73 | 4.15 | 4.00 | 3.16 | −0.03 | −0.03 | −0.04 |

*3.2. Match and Mismatch between ESs Supply and Demand*

The *ESDR* of water yield was high in the northwest and southeast, low in the middle parts of the BLJW, and a water-yield deficit was found in the Zhouqu-Wudu section of the BLJW. The main reason is that the precipitation distribution of the BLJW is higher in the northwest and southeast and lower in the middle, which leads to the low water-yield supply in the middle parts. At the same time, a larger population resides in the central and eastern BLJW, with high water demand. The water-yield supply and demand primarily belong to the high supply with low demand (northwest of the BLJW) and low supply with high demand clusters (east of the BLJW) (Figure 4A), with a decreasing trend (18.12%, 14.42%, 13.28%, and 14.01%, 12.86%, 11.49%, respectively). The high-supply high-demand clusters increased significantly (1.52%, 3.27%, and 4.53%, respectively), primarily in southern Wudu. The low-supply low-demand clusters were mainly in the north of Diebu, and the area proportion decreased first and then increased (11.59%, 8.95%, and

12.95%, respectively) (Figure 5A). The spatial-match patterns of soil supply and demand in the BLJW showed less variation from 2002 to 2018. The *ESDR* of soil conservation was high in the south and west of the BLJW, with a surplus situation, while the low-value areas were scattered, mainly in the north of Diebu, the northern part of Tanchang, eastern Wudu, and the north of Wenxian. The *ESDR* pattern of soil conservation was mainly affected by factors such as precipitation, vegetation cover, terrain, and soil type. The deficit areas were primarily distributed in the lithoid mountainous regions in the northwest, cultivated land, and its surroundings (Figure 4B). As to the spatial-matching type of the supply and demand, the dominant type of soil conservation was low supply with low demand, with an increasing trend (11.10%, 12.32%, and 12.46%, respectively), mainly concentrated in the northwest of Tanchang, the subareas between Gongba River and Bailongjiang, and the eastern parts of the Zhouqu-Wudu sections of the Bailongjiang (Figure 5B). The spatial patterns of food-provision *ESDR* varied less from 2002 to 2018, and the deficit was evident in the eastern BLJW, while the surplus subareas were scattered. Food provision maintains a balance between supply and demand in the western BLJW, and the shortage of food supplies was concentrated in the densely populated settlements (Figure 4C). As to the spatial-matching type of supply and demand, the dominant type of food provision was high supply with high demand (8.78%, 7.88%, and 7.97%, respectively), mainly distributed in the eastern parts of the Tanchang-Zhouqu-Wudu section of the BLJW and northern Wenxian, with better agricultural conditions and dense populations (Figure 5C).

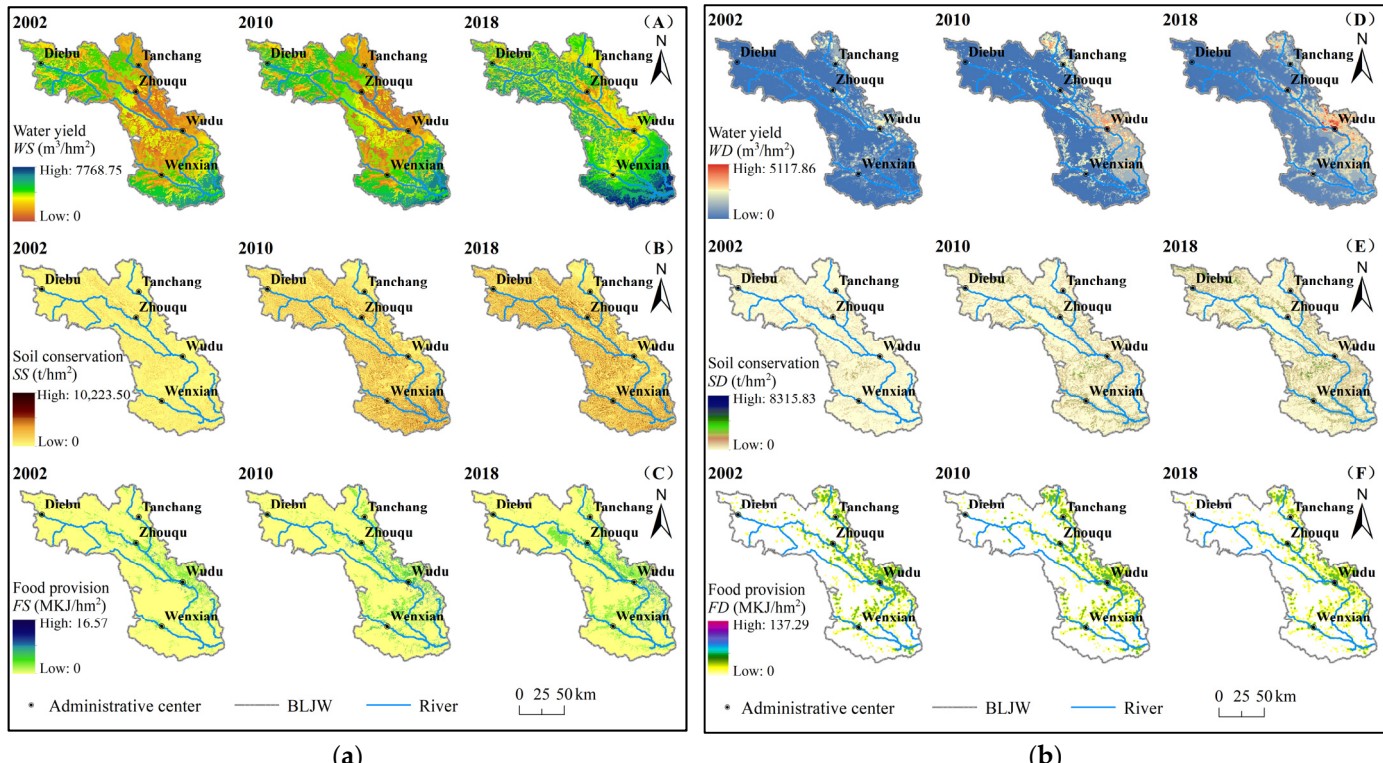

**Figure 3.** Spatial distributions of the ESs supply (**a**) and demand (**b**) in the BLJW from 2002 to 2018. Note: series (**A**) shows the water supply from 2002 to 2018, series (**B**) shows the soil-conservation supply from 2002 to 2018, series (**C**) shows the food supply from 2002 to 2018, series (**D**) shows the water demand from 2002 to 2018, series (**E**) shows the soil-conservation demand from 2002 to 2018, series (**F**) shows the food demand from 2002 to 2018.

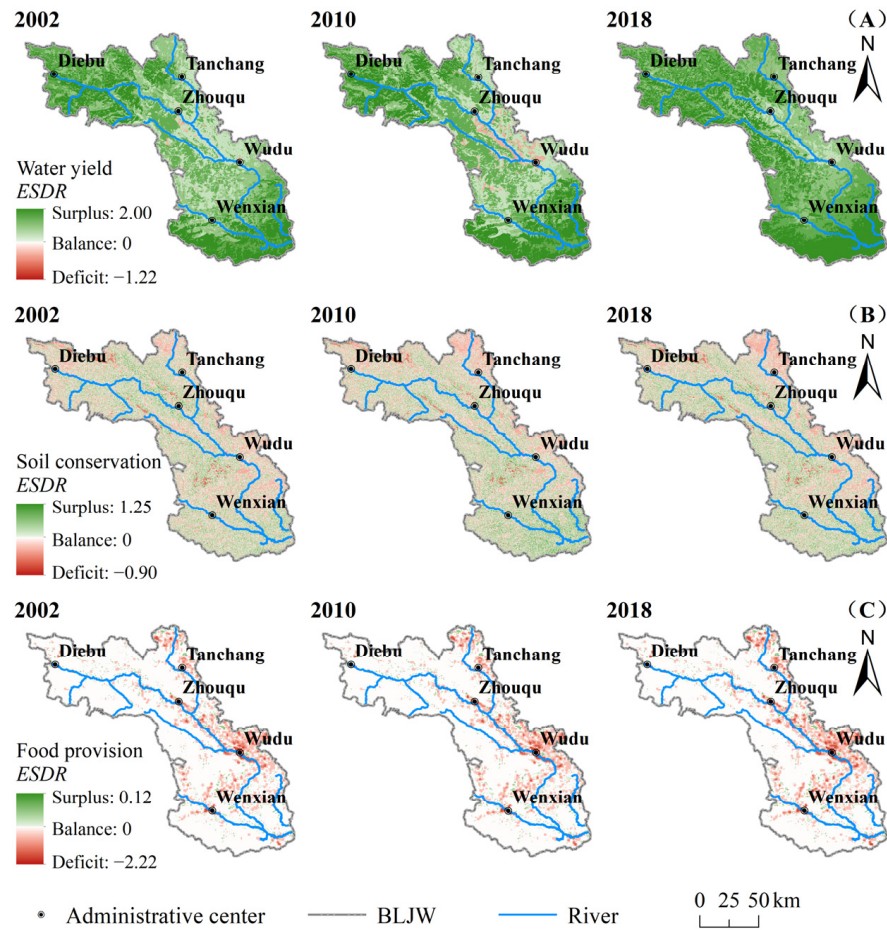

**Figure 4.** Spatial pattern of the supply–demand ratio of the ESs in the BLJW from 2002 to 2018. Note: series (**A**) shows the *ESDR* of water yield from 2002 to 2018, series (**B**) shows the *ESDR* of soil conservation from 2002 to 2018, series (**C**) shows the *ESDR* of food provision from 2002 to 2018.

*3.3. ESs Trade-Off and Synergy*

Spearman's correlation analysis determined the correlation between the two variables. In order to better reveal the ESs tradeoffs and synergies, after a series of grid size tests by the Fishnets method, a 5-km grid was selected to obtain the mean ESs with the most significant correlation. SPSS was used for correlation analysis. The relationship between water yield (WY) and soil conservation (SC) (WY-SC, hereafter) in the BLJW was one of synergy (0.291 **, 0.463 **, and 0.379 **) (Table 2). The trade-off intensity of WY-SC was higher in the southeast and northwestern BLJW and lower in the middle of the BLJW, with an increasing trend from 2002 to 2018, especially in the middle parts of the BLJW. We analyzed the trade-off benefits bias at the township scale to reveal the trade-off relationship. The soil-conservation function is more advantageous in most areas of the BLJW in the WY-SC trade-off relationship, and the water yield was more dominant in the northwest, southeast, and northeast. The relationship between water yield and food provision (WY-FP, hereafter) and that of soil conservation and food provision (SC-FP, hereafter) were both trade-offs (−0.396 **, −0.351 **, and −0.293 **; −0.261 **, −0.129 **, and −0.180 **, respectively). The spatial and temporal performance of the WY-FP trade-off intensity was similar to that of the WY-SC. Most areas of the BLJW are mainly biased toward the water-yield benefits. The food-provision function areas are concentrated in the eastern parts of the Zhouqu-Wudu section of Bailongjiang. The trade-off intensity of SC-FP was low, with an average value of 0.03, and the soil-conservation capacity was better in most areas of the BLJW. The trade-off intensity of SC-FP was higher in the west and south of the BLJW, while it is lower in the northeast and the northwest BLJW and in the Wudu-

Wenxian section of Bailongjiang. Regarding the trade-off relationships between SC-FP, the subregions biased toward the food-provision benefits were primarily located in the eastern part of the Zhouqu-Wudu section of Bailongjiang, indicating that the middle and east of the BLJW were the main subregions for agricultural production and food provision (Figure 6).

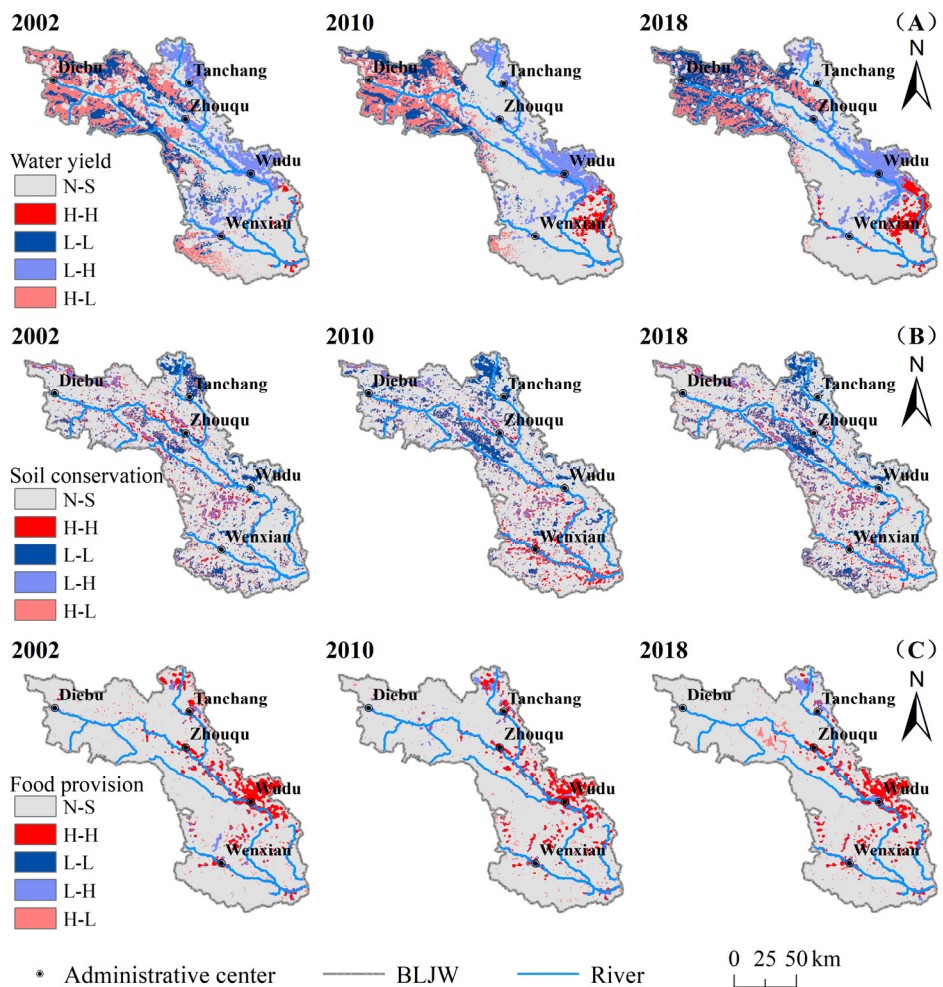

**Figure 5.** Cluster maps for the ESs supply–demand in the BLJW from 2002 to 2018. Note: H–H: high supply–high demand clustering, L–L: low supply–low demand clustering, H–L: high supply–low demand clustering, L–H: low supply–high demand clustering, and N-S: not significant relationship. Series (**A**) shows the cluster of water yield supply and demand from 2002 to 2018, series (**B**) shows the cluster of soil conservation supply and demand from 2002 to 2018, series (**C**) shows the cluster of food provision supply and demand from 2002 to 2018.

**Table 2.** Correlation coefficients between ES pairs in the BLJW in Gansu from 2002 to 2018.

| Year | Index | WY-SC | WY-FP | FP-SC |
|------|-------|-------|-------|-------|
| | *RMSD* range | [0–0.707] | [0–0.707] | [0–0.686] |
| 2002 | *RMSD* mean | 0.302 | 0.326 | 0.035 |
| | Correlation coefficients | 0.291 ** | −0.396 ** | −0.261 ** |
| | *RMSD* range | [0–0.705] | [0–0.707] | [0–0.703] |
| 2010 | *RMSD* mean | 0.308 | 0.329 | 0.033 |
| | Correlation coefficients | 0.463 ** | −0.351 ** | −0.129 ** |

**Table 2.** *Cont.*

| Year | Index | WY-SC | WY-FP | FP-SC |
|---|---|---|---|---|
| | *RMSD* range | [0–0.707] | [0–0.707] | [0–0.707] |
| 2018 | *RMSD* mean | 0.366 | 0.391 | 0.036 |
| | Correlation coefficients | 0.379 ** | −0.293 ** | −0.180 ** |

Note: ** means *p* < 0.01.

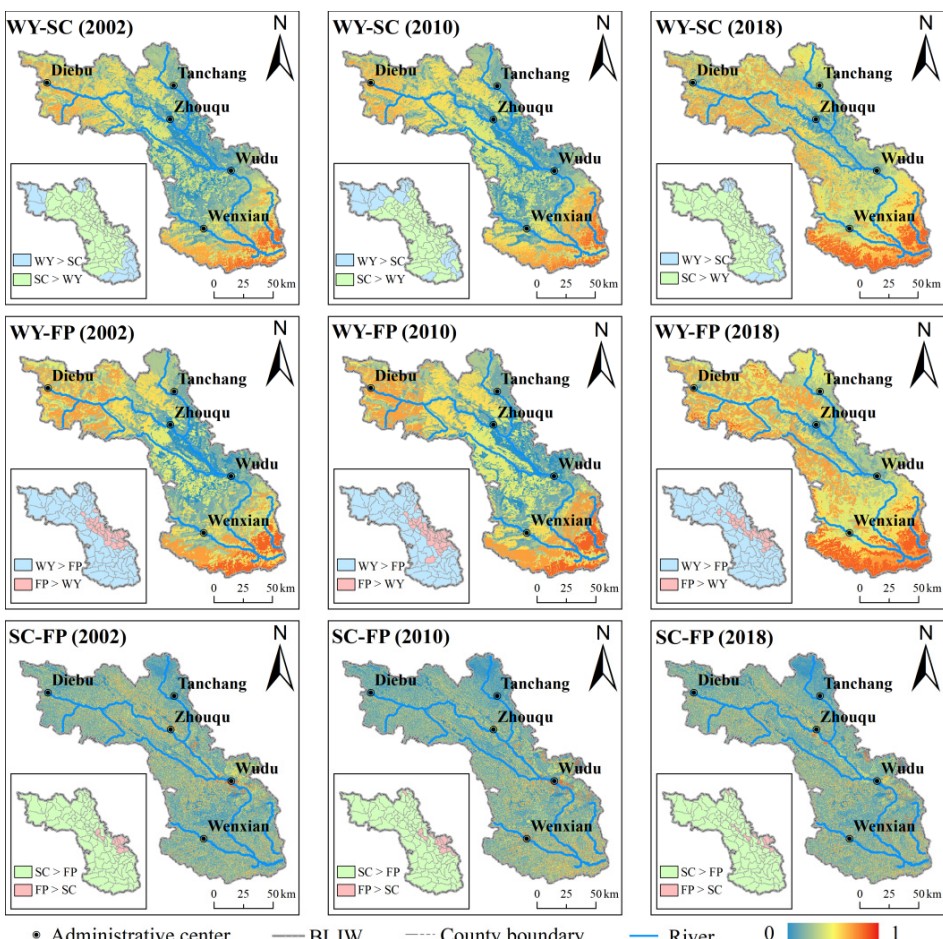

**Figure 6.** Map of the ESs trade-off intensity in the BLJW from 2002 to 2018.

### 3.4. The Relationship between the Trade-Off Intensity and the Supply–Demand Ratio of the Watershed ESs

There is a synergistic relationship between the WY-SC and a trade-off relationship between WY-FP and SC-FP in the BLJW. A trade-off is more critical than synergy in balancing the allocation of natural resources. Therefore, we chose WY-FP and SC-FP to analyze the relationship between the trade-off intensity and supply–demand ratio of ESs using the quantile regression method of the SPSS. Quantile regression does not require a normal distribution and enables the study of relationships between the independent variable and different conditional quantiles of the dependent variable [84,85]. According to Table 3, we analyzed the relationship between the trade-off intensity (*RMSD*) and different quantiles of the *ESDR* under different scenarios in the trade-off relationship of WY-FP and SC-FP. The scenarios are as follows: water yield increases while food provision decreases (WY is more advantageous, hereafter Area$_{WY>FP}$), food provision increases while water yield decreases (FP is more beneficial, hereafter Area$_{FP>WY}$), soil conservation increases while food provision decreases (SC is more advantageous, hereafter as Area$_{SC>FP}$), and

food provision increases while soil conservation decreases (FP is more beneficial, hereafter Area$_{FP>SC}$).

**Table 3.** Quantile regression between the trade-offs intensities and supply–demand ratio.

| *ESDR* Quantiles | **Area$_{WY>FP}$** | | **Area$_{FP>WY}$** | | **Area$_{SC>FP}$** | | **Area$_{FP>SC}$** | |
|---|---|---|---|---|---|---|---|---|
| | *ESDR$_{WY}$* | *ESDR$_{FP}$* | *ESDR$_{WY}$* | *ESDR$_{FP}$* | *ESDR$_{SC}$* | *ESDR$_{FP}$* | *ESDR$_{SC}$* | *ESDR$_{FP}$* |
| 10% | 0.329 ** | 0.042 | −1.894 | −0.351 | −0.768 ** | 0.340 | −0.329 | 0.469 |
| 30% | −0.028 | 0.102 ** | −2.685 ** | −0.098 | −0.626 ** | 0.621 * | −0.238 | 0.112 |
| 50% | −0.277 ** | 0.056 ** | −2.375 ** | −0.025 | −0.415 ** | 0.371 ** | −0.073 | 0.001 |
| 70% | −0.434 ** | 0.005 | −1.885 ** | 0.000 | −0.286 ** | 0.049 | 0.166 | 0.000 |
| 90% | −1.051 ** | 0.000 | −1.444 ** | 0.000 | −0.326 ** | 0.003 | −0.232 | 0.001 |

Note: * means $0.01 < p < 0.05$; ** means $p \leq 0.01$.

The correlation coefficient between the *RMSD* and *ESDR* of WY and SC (*ESDR$_{WY}$* and *ESDR$_{SC}$*, hereafter) is negative. The *ESDR$_{WY}$* and *ESDR$_{SC}$* decreased gradually with the increase in *RMSD*. The relationship between the *RMSD* of WY-FP and SC-FP and the *ESDR* of food provision (*ESDR$_{FP}$*, hereafter) is insignificant (Table 3). In the Area$_{WY>FP}$, the *ESDR$_{WY}$* has a more robust negative correlation response to the *RMSD* of WY-FP when the *ESDR$_{WY}$* is high. The *ESDR$_{WY}$* has a more robust positive correlation response to the *RMSD* of WY-FP when the *ESDR$_{WY}$* is small (large contradiction between water-yield supply and demand). Thus, increasing the *RMSD* of WY-FP is conducive to alleviating the contradiction between water-yield supply and demand. In the Area$_{FP>WY}$, the negative response is insignificant when the *ESDR$_{WY}$* is small, which indicates that when the contradiction between water-yield supply and demand reaches a high level, the conflict between local water and food supply is no longer critical. Adjusting the WY-FP trade-off relationship is no longer essential to improve the contradiction between water-yield supply and demand. As to the Area$_{SC>FP}$, the negative response of *ESDR$_{SC}$* to the *RMSD* of SC-FP is more robust when the *ESDR$_{SC}$* is relatively small (large contradiction between soil-conservation supply and demand). Therefore, reducing the *RMSD* of SC-FP is conducive to alleviating the contradiction between soil-conservation supply and demand. In the Area$_{FP>SC}$, the correlation coefficient between the *RMSD* values of SC-FP and *ESDR$_{SC}$* failed to pass the significance test, and no relationship was apparent.

## 4. Discussion

### 4.1. Implications of ESs Supply and Demand Relationships

Considering the comprehensive natural conditions of the BLJW and the related studies [58,68,74,86], we analyzed the effects of the main influencing factors on the ESs supply and demand, such as land-use proportion, temperature, precipitation, altitude, slope, vegetation coverage, economic density, and population density (Table 4). Under the scenarios of Area$_{WY>FP}$ and Area$_{SC>FP}$, socio-economic factors such as economic density, population density, the proportion of cultivated land, and the proportion of construction land can have a positive impact on the supply–demand relationships of water yield and soil conservation, which indicates that appropriate human activities could improve the water-yield and soil-conservation *ESDR*s in the subareas with less human impact. Both scenarios are located in areas with fewer farming activities, and the correlation between food supply and demand and various effects factors is low. In the Area$_{FP>WY}$ scenario, the *ESDR$_{WY}$* is mainly negatively related to the cultivated area. The contradiction between water-yield supply and demand is more serious when there is a large farmland proportion. The correlation between the food supply–demand relationship and various effect factors is weak. In the Area$_{FP>SC}$ scenario, the *ESDR$_{FP}$* positively correlates with the temperature, precipitation, economy density, population density, and farmland proportion. The contradiction between the supply and demand of soil conservation has little correlation with various effect factors,

indicating that the soil-conservation situation is affected by cultivation activities, which makes the supply and demand situation more complicated. In addition, it was found that the dominant influencing factors between the ESs supply–demand relationship and the trade-off intensity have overlaps, among which land use is the main common factor and the internal reason for their correlation. Forestland and grassland account for the most significant proportion in the scenarios of Area$_{WY>FP}$ and Area$_{SC>FP}$ (46.90% and 36.62%, 46.22%, and 36.36%, respectively). Farmland and construction land account for the most significant proportion in the scenarios of Area$_{FP>WY}$ and Area$_{FP>SC}$ (44.19% and 2.04%, 46.80%, and 3.03%, respectively). The proportion of farmland and construction land significantly affects the supply–demand ratio and trade-off intensity, but the impact direction was inconsistent. Therefore, it is necessary to strengthen the effective use of land resources to improve the ESs function of the BLJW and reduce the ESs supply–demand risks [87].

**Table 4.** Effects of factors on trade-offs and supply–demand ratio.

| Social-Ecological Factors | Area$_{WY>FP}$ | | | Area$_{FP>WY}$ | | | Area$_{SC>FP}$ | | | Area$_{FP>SC}$ | | |
|---|---|---|---|---|---|---|---|---|---|---|---|---|
| | **WY** | **FP** | *RMSD* | **WY** | **FP** | *RMSD* | **SC** | **FP** | *RMSD* | **SC** | **FP** | *RMSD* |
| Temperature | 0.077 * | 0.037 | 0.260 ** | 0.162 | 0.017 | 0.184 | −0.071* | 0.017 | 0.264 ** | 0.211 | 0.523 ** | 0.307 |
| Precipitation | −0.120 ** | 0.021 | 0.575 ** | −0.238 | −0.028 | 0.364 ** | −0.130 ** | 0.017 | 0.258 ** | 0.005 | 0.434 ** | 0.371 ** |
| Elevation | −0.094 * | −0.041 | −0.187 ** | −0.188 | −0.020 | −0.136 | 0.068 | −0.014 | −0.237 ** | −0.206 | −0.017 | 0.068 |
| Slope | −0.038 | 0.083 | 0.413 ** | −0.060 | −0.002 | −0.269 ** | −0.254 ** | 0.037 | 0.697 ** | 0.226 | 0.420 | 0.354 * |
| NDVI | 0.031 | 0.113 ** | 0.532 ** | −0.074 | 0.013 | 0.005 | −0.210 ** | −0.016 | 0.528 ** | 0.010 | −0.151 | 0.274 |
| Economic density | 0.083 * | −0.005 | 0.184 ** | 0.082 | −0.055 | 0.369 ** | 0.072 * | 0.055 | 0.172 ** | −0.028 | 0.650 ** | 0.510 ** |
| Population density | 0.106 ** | −0.049 | −0.192 ** | 0.006 | −0.084 | −0.126 | 0.150 ** | −0.009 | −0.149 ** | 0.050 | 0.471 ** | 0.341 * |
| Farmland Proportion | 0.103 ** | −0.051 | −0.215 ** | 0.212 * | 0.005 | −0.229 * | 0.140 * | −0.037 | −0.156 ** | −0.055 | 0.360 * | 0.066 |
| Forestland Proportion | −0.039 | 0.126 ** | 0.339 ** | −0.044 | −0.061 | −0.186 | −0.194 ** | 0.039 | 0.419 ** | 0.007 | 0.218 | 0.255 |
| Grassland Proportion | 0.085 * | −0.074 * | −0.215 ** | −0.173 | 0.023 | 0.321 ** | 0.061 | −0.015 | −0.065 | 0.195 | 0.309 | 0.278 |
| Water area Proportion | 0.092 * | −0.006 | −0.076 * | 0.177 | 0.049 | −0.269 ** | −0.007 | 0.007 | −0.040 | 0.086 | 0.141 | 0.022 |
| Construction land Proportion | 0.143 ** | 0.037 | −0.153 ** | 0.032 | −0.184 | −0.033 | 0.085 * | −0.013 | −0.126 ** | 0.104 | 0.214 | 0.018 |
| Unused land Proportion | −0.042 | −0.062 | −0.111 ** | −0.199 * | −0.087 | −0.081 | −0.002 | −0.036 | −0.050 | 0.142 | 0.054 | −0.079 |

Note: * means $0.01 < p < 0.05$; ** means $p \leq 0.01$.

### 4.2. Governance Suggestions Based on ESs Supply and Demand Changes

Referring to various effect factors and the relationship between the trade-off intensity and the supply–demand situation, it is more conducive to putting forward relevant suggestions to improve the ESs function and reduce the ESs supply and demand risk. The eastern parts of the BLJW have a high intensity of land use, mainly belonging to cultivated land and population gathering subareas, and various ESs are prone to be in short supply. Both the Area$_{FP>WY}$ and Area$_{FP>SC}$ scenarios are concentrated in the east of the Zhouqu-Wudu section of the Bailongjiang. For areas where the water-yield supply is in a deficit, the water demand mainly depends on water transfer from other places. The slope cropland, wasteland, deforestation, and degraded grassland should be effectively managed, as well as strengthening the construction and application of flood control and storage and drainage projects [58]. The subareas of soil-conservation and food-provision supply deficit have many overlaps. There is a trade-off relationship between soil conservation and food production, so choosing according to the actual local condition is necessary [56]. Agricultural land management should be conducted according to the quality classification of cultivated land in the province. It is inappropriate to increase grain output by blindly expanding cultivated land. The food-supply deficit will reduce via low-efficiency farmland and agricultural

technology. The southern BLJW has a humid climate with abundant precipitation and high vegetation coverage, an important water-source protection area with a high water-yield supply. The western BLJW has more forestland with low human disturbance and many nature reserves, which are essential ecological functional areas for soil conservation. Most of the food production is from animal husbandry with lower populations. The above subareas are all under the scenarios of Area$_{WY>FP}$ and Area$_{SC>FP}$, more efforts should be paid to comprehensively protect the forests and grasslands, strengthen the ecological management of natural reserves, and properly develop the environmental tourism-based economy [58,68,86,88].

*4.3. Limitations and Prospects*

This paper analyzed the spatial–temporal variation in the ESs supply and demand, trade-off/synergy relationship, and spatial-matching characteristics from 2002 to 2018 in the BLJW. This study made a unified comparison between the ESs supply and demand by localizing the model parameters. Thus, it can better verify the results by combining the complex socio-ecological conditions in the BLJW. However, due to the complexity of the ESs formation mechanism and limitations of the models and data, further integration of various evaluation indicators and techniques remains to be carried out for future improvement. Due to the actual situation of the BLJW, this study analyzed the balance between water-yield, soil-conservation, and food-provision supply and demand. In the future, the evaluation of other ESs should be considered. Moreover, the accuracy and comprehensiveness of the assessment data should be improved to further study the dynamic relationship. The systematic understanding of the spatiotemporal scale effect of the supply and demand relationship of ESs has a guiding role [89]. More attention should be paid to long-term statistical analysis at the county and township scales for ecological management. It is necessary to adjust the relationship between the ESs supply and demand to increase the supply of multiple ESs and to improve ESs supply and demand. We must identify the formation mechanism of the interaction between different types of ESs or simultaneous influencing factor associations [25]. In addition, the influencing factors of the intrinsic relationship between trade-off intensity and supply–demand ratio are relatively complex. This study analyzed the effect factors on the ESs supply and demand; the interaction and correlation mechanisms between ESs still need to be deepened. Although this study has proposed suggestions for watershed ESs management based on the ESs supply–demand trade-off and synergy, the practicability of management suggestions needs to be further improved due to the management difficulty of different administrative divisions. We can gradually establish a governance mechanism of ESs supply and demand by zoning and grading by comparing the types and intensity differences of trade-offs and synergies between different ESs supply and demand. We must coordinate the supply and demand of different kinds of ESs in various regions, alleviate the imbalance between ESs supply and demand, and maximize the benefits of various ecosystem services.

**5. Conclusions**

In this study, based on a quantitative analysis of ESs supply and demand changes from 2002 to 2018 in the BLJW, a comprehensive research framework of ESs assessment was developed by coupling the ESs trade-offs and supply–demand dynamics. ESs supply and demand characteristics in the BLJW are spatially heterogeneous. The differences in the spatial structure of ESs supply make their corresponding ecological functions different, which are closely related to the local natural conditions. From 2002 to 2018, the water-yield, soil-conservation, and food-provision supply showed an increasing trend (increased by 33.75%, 102.65%, and 94.65%, respectively). The spatial characteristics of water and food demand are closely related to population density. From 2002 to 2018, the water and soil conservation demand increased by 133.33% and 148.78%, while food demand decreased by 23.86%. As to the spatial characteristics of the supply and demand match type, there was an apparent surplus in water yield, the soil conservation was mainly balanced, and the food

provision showed a deficit along the eastern part of the BLJW. The relationship between water yield and soil conservation was synergetic, while the water yield–food provision and soil conservation–food provision showed trade-offs. Coupled ESs supply–demand and tradeoff showed that with the increase in trade-off intensity, the *ESDR* of water yield and soil conservation showed a downtrend, while the change in the food-provision *ESDR* was not significant. The composition of land use was significantly different for different trade-off benefit areas. Based on the coupling analysis of ESs supply–demand and trade-off relationships, corresponding land-use management suggestions were put forward with a combination of local ecological and social-economic characteristics.

**Author Contributions:** Conceptualization, Y.Z. and J.G.; methodology, Y.Z. and Y.W.; validation, Y.Z. and Y.W.; formal analysis, Y.Z. and J.S.; writing—original draft preparation, Y.Z.; writing—review and editing, Z.H. and J.G.; visualization, Y.Z., Y.W. and J.S.; supervision, J.G.; funding acquisition, J.G. All authors have read and agreed to the published version of the manuscript.

**Funding:** This research was funded by the Second Tibetan Plateau Scientific Expedition and Research (Grant No. 2019QZKK0603) and the National Natural Science Foundation of China (Grant No. 42171090).

**Institutional Review Board Statement:** Not applicable.

**Data Availability Statement:** Data are available from the authors upon reasonable request, as the data are required for further use.

**Conflicts of Interest:** The authors declare no conflict of interest.

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
