# Peer review of "Spatiotemporal Changes in the Watershed Ecosystem Services Supply and Demand Relationships in the Eastern Margin of the Qinghai-Tibetan Plateau"

_diversity, doi:10.3390/d15040551_

Round 1
Reviewer 1 Report
This paper studies the ESs supply and demand relationships and the ESs trade-offs and synergies in Bailongjiang Watershed in western China. Through the analysis of three ES the authors try to understand the supply and demand spatial distribution and dynamic relationship of these ESs with the aim of formulating management policies. Although the subject of the work is well developed in the international literature, each new study is useful in understanding the relationships between the ESs, and the work contributes to new knowledge on these issues.
Although the paper has very good documentation of current research in the introduction, it uses a small number of ES to formulate management policies, especially since the area is predominantly forest ESs and as the authors report also in Lines 150-151" Due to the intensive human activities and complex natural environment, the fragile ecology is one of the high-incidence places with frequent geo-disasters such as debris flow and landslides in China. Therefore, coordinating the supply and demand of ESs is vital for ecological protection and human welfare”. This is not problematic for the integrity of the study, as two of the three ES that have been used contribute to the formulation of local ESs policy management and protection of land, but the possible use of other ES such as Regulating ES: Carbon sequestration and climate regulation, or Provisioning services: food and raw materials from forest areas, should give a more complete picture of the critical ecological functions of water and soil conservation of the area. Why did the authors only use these 3 ESs although they had already analysed satellite imagery for the entire study area?
Comments
There are some major comments on this work.
1. In the Results section, what is presented could be reduced considerably in length and most of the information could be presented in tables and charts. At the same time, most of the text of the Discussion Section is essentially results (e.g. the analysis for Quantile regression). The Results and Discussion sections should be configured accordingly, where in the latter there should be an appropriate discussion and in relation to the bibliography (as has been very correctly developed in the Introduction).
2. Line 156. 2.2. Data sources and processing. Important information for the analyses or referral to a corresponding paper is missing. The authors report, that they used Landsat (30m resolution) and MODIS whose minimum resolution is 250m or 500m. Has Landsat used them in their study and how many years the analysis was done? With what methodology of analysis in ENVI the Landsat was processed (the authors reported in Line 161 "accuracy was estimated over 86%". How did MODIS values downscaling to the scale of Landsat and DEM?”
Minor comment
1. There is an incorrect citation in the literature on Line 539. Guo, C.Q.; Xu, X.B.; Shu, Q. A review on the assessment methods of supply and demand of ecosystem services. Journal of Ecology 2020, 39, 2086–2096. This paper has not been published in the journal cited but in the Chinese Journal of Ecology 2020, 39, 2086–2096. This needs a lot of attention and should be mentioned, in parentheses, all the articles where the language of writing is Chinese of the total 89 articles.
2. Line 155. Fig 2. It needs to include the relative position of the area in the China country. There should also be a table with the relative proportion of the types of coverage presented in the diagram.
3. The authors quote Lines 202- 203 "There is a significant linear relationship between NDVI and food production. Therefore, the food supply (FS) is calculated by corresponding to NDVI”. According to the diagrams, the analysis was carried out in the rural zone and not in the forest area. This should be clearly reflected in the text.
4. Is the ecological supply-demand ratio (ESDR) equation, correct? Is the term ED min correct?
Author Response
Dear Editor and reviewers,
Many thanks for your consideration and the valuable and detailed suggestions on our manuscript Ref: Diversity-2289862. This manuscript has been revised carefully according to the comments and constructive suggestions provided by you and the unknown reviewers. All the changes have been marked in an annotated version of the revised manuscript (submission item “Revised manuscript with track changes”), and the detailed revision points have been listed in " Responses to editor and reviewers".
Should you have any questions or wish to communicate by email, I will be pleased to do so. We hope that this revision could meet your satisfaction and provides an acceptable manuscript. Please kindly have a check. Many thanks for your nice and consideration.
Here are brief explanations of the questions and suggestions from reviewers and editors, pleased to find them below.
By the way, here, the Q & C represents Question and Comments, and the A represents the Answers and explanations.
Yours Sincerely,
Corresponding author
Jie Gong
Reviewer 1
Q & C-1: This paper studies the ESs supply and demand relationships and the ESs trade-offs and synergies in Bailongjiang Watershed in western China. Through the analysis of three ESs the authors try to understand the supply and demand spatial distribution and dynamic relationship of these ESs with the aim of formulating management policies. Although the subject of the work is well developed in the international literature, each new study is useful in understanding the relationships between the ESs, and the work contributes to new knowledge on these issues.
Although the paper has very good documentation of current research in the introduction, it uses a small number of ES to formulate management policies, especially since the area is predominantly forest ESs and as the authors report also in Lines 150-151" Due to the intensive human activities and complex natural environment, the fragile ecology is one of the high-incidence places with frequent geo-disasters such as debris flow and landslides in China. Therefore, coordinating the supply and demand of ESs is vital for ecological protection and human welfare”. This is not problematic for the integrity of the study, as two of the three ES that have been used contribute to the formulation of local ESs policy management and protection of land, but the possible use of other ES such as Regulating ES: Carbon sequestration and climate regulation, or Provisioning services: food and raw materials from forest areas, should give a more complete picture of the critical ecological functions of water and soil conservation of the area. Why did the authors only use these 3 ESs although they had already analyzed satellite imagery for the entire study area?
A: Thank you very much for your comments and nice suggestions.
The study area, the Bailogjiang Watershed (BLJW), is experiencing intensive human activities, significant expansion of population, and urbanized land due to the rapid economic development via image interpretation and field investigation. The primary source of urban expansion caused the reduction of cultivated land, as well as the shortage of food production. Therefore, it is urgent to carry out research on food supply and demand matching. Furthermore, the BLJW is a vital ecological function area of water and soil conservation in the Eastern Margin of the Qinghai-Tibetan Plateau. However, the water yield and soil conservation capacity in the BLJW are unbalanced distributed. Besides, frequent natural disasters such as landslides and debris flows have led to severe water and soil loss. Therefore, it is urgently needed to improve ecosystem service and the primary material conditions for human life. Thus, three crucial ecosystem service (ESs) types, including water yield, soil conservation, and food provision services, were selected and assessed to show the spatiotemporal change and its supply and demand relationships in the BLJW.
More words on the research progress of ecosystem services and the study area have been added, especially in the Introduction and Study area and the Discussion parts. We will continue to study other ESs types and their supply and demand relationships in the future. Pleased to find the details from the introduction, study area, and discussion parts (4.3 Limitations and prospects).
Q & C-2: In the Results section, what is presented could be reduced considerably in length and most of the information could be presented in tables and charts. At the same time, most of the text of the Discussion Section is essentially results. The Results and Discussion sections should be configured accordingly, where in the latter there should be an appropriate discussion in relation to the bibliography.
A: Thank you very much for your comments and nice suggestions.
We have adjusted the content of the result and discussion. As suggested by other reviewers, ecosystem services are fluid. The BLJW exists for allocation and transportation in the ecosystem services supply, such as water supply and food supply. Therefore, it is necessary to show the distribution characteristics of supply, demand, and the relationship between supply and demand simultaneously in space, which needs to increase the correlation analysis and elaboration. A simple tabular presentation can be misleading in understanding supply and demand.
Q & C-3: Line 156. 2.2. Data sources and processing. Important information for the analyses or referral to a corresponding paper is missing.
A: Thank you very much for your comments and nice suggestions.
More words on the data source, data pre-processing methods, and other information have been added for more details. Pleased to find the details from section “2.3 Data sources and processing”.
Q & C-4: The ninth citation source in the literature is incorrect. This needs a lot of attention and should be mentioned, in parentheses, in all the articles where the language of writing is Chinese of the total 89 articles.
A: Thank you very much for your suggestions. The references had been checked and revised accordingly. Pleased to find them in the reference parts.
Q & C-5: Fig 2. It needs to include the relative position of the area in the China country. There should also be a table with the relative proportion of the types of coverage presented in the diagram.
A: Thanks for your comments and suggestions. Figure 2 has been modified and revised accordingly. The location of the study area in China (Figure 2a) and the proportion of various land uses (Figure 2b) have been added for more clarity. Pleased to find the details from the new Figure 1 in the section “Study area and methods”.
Q & C-6: Section 2.3.3. According to the diagrams, the analysis was carried out in the rural zone and not in the forest area. This should be clearly reflected in the text.
A: Thank you very much for your comments and nice suggestion.
The calculation methods of food supply had been checked and revised accordingly for more reasonable. Liu et al. (2019) reported that “there is a significant linear relationship between farmland NDVI and food production.” Therefore, the NDVI of the cultivation land is used to calculate the food supply (FS). More details can be found in the revised manuscript and reference Liu et al. (2019) (Ref. No. 76).
Q & C-7: Is the ecological supply-demand ratio (ESDR) equation correct? Is the term ED min correct?
A: Thank you very much for your comments.
We are sorry for a clerical error on the ecological supply-demand ratio (ESDR). The ESDR equation has been revised accordingly. Pleased to find the correct equation from the method parts in the manuscript.

Reviewer 2 Report
I found this article actual and interesting. It is based on solid data, it applies established methods for analysing ES supply and demand, and the results are well presented and critically discussed. All declared aims of the research were fulfilled.
Introduction is quite long, especially the second paragraph (lines54-98). I suggest shortening it. Authors may consider moving Fig. 1 into Study and methods, although I see why they placed it at the end of Introduction.
Fig. 2 - I suggest to show the location of the study are within China to help reader to locate it. Maps are rather small and hard to read. It applies to other maps as well. Land use classes "Construction land" and "Unused land" are rather odd terms _ I suggest to use Built-up land or Urbanised land or explain in the text what is meant by Unused land.
I suggest elaborating a bit more on spatial relations between ES supply and demand and their match or mismatch as for certain ES location of supply and demand may be identical (soil prevention) whilst with others as water yield and food provision the location of demand is often elsewhere (where the population is located). For this reason, a simple spatial overlay of the supply and the demand may be misleading.
Author Response
Dear editor and reviewers,
Many thanks for your considerations and the valuable and detailed suggestions on our manuscript Ref: Diversity-2289862. This manuscript has been revised carefully according to the comments and constructive suggestions provided by you and the unknown reviewers. All the changes have been marked in an annotated version of the revised manuscript (submission item “Revised manuscript with track changes”), and the detailed revision points have been listed in " Responses to editor and reviewers".
Should you have any questions, or wish to communicate by email, I will be pleased to do so. We hope that this revision could meet your satisfaction and provide an acceptable manuscript. Please kindly have a check. Many thanks for your nice and considerations.
Here are the brief explanation of the questions and suggestions from reviewers and editors, pleased to find them as below.
By the way, here, the Q & C represents Question and Comments, and the A represent the Answers and explanations.
Yours Sincerely,
Corresponding author
Jie Gong
Reviewer 2
Q & C-1: The introduction is quite long, especially the second paragraph (lines 54-98). Suggest shortening it. Authors may consider moving Fig. 1 into the Study and methods.
A: Thank you very much for your comments and nice suggestions.
The introduction part has been reorganized, shortened, and revised accordingly for more condense and clarity. Some sentences have been removed to shorten the paragraph and introduction parts. The research aims and framework (Fig. 1) has been moved to Section 2 “Study area and methods”. Pleased to find the details in the revised manuscript.
Q & C-2: Fig. 2 suggested showing the location of the study within China to help readers to locate it. Maps are rather small and hard to read. Land use classes "Construction land" and "Unused land" are rather odd terms—which suggested using Built-up land or Urbanised land or explaining in the text what is meant by Unused land.
A: Thank you very much for your comments and nice suggestions.
Figure 2 (now Figure 1) has been revised accordingly for more clarity. The location of the study area in China (Figure 2a) and the proportion of various land uses (Figure 2b) have been added for more clarity. Pleased to find the details in the new Figure 1 in the section “Study area and methods”.
The legend of Fig.2 has been revised accordingly, too. The “Construction land” has been changed to “Built-up land,” the “Unused land” has been changed to “unexploited land,” which refers to land covers such as bare land and bare rock, other than agricultural and construction uses, according to the common land use classification in China.
Pleased to find the details in the revised manuscript.
Q & C-3: Elaborating a bit more on spatial relations between ES supply and demand and their match or mismatch.
A: Thank you very much for your comments and nice suggestion.
More words have been added to analyze the spatial relationships between ES supply and demand and their match or mismatch, especially in section 3.4, “The relationship between the trade-off intensity and the supply-demand ratio of the watershed ESs”, section 4.2, “Governance suggestions based on ES supply and demand changes”, and section 4.3 “Limitations and prospects”. Pleased to find the details in the revised manuscript.
Round 2
Reviewer 1 Report
Thanks to the authors for the revised MS. In this new version, I agree that it can be published.